# An Embedded Portable Lightweight Platform for Real-Time Early Smoke Detection

**DOI:** 10.3390/s22124655

**Published:** 2022-06-20

**Authors:** Bowen Liu, Bingjian Sun, Pengle Cheng, Ying Huang

**Affiliations:** 1School of Technology, Beijing Forestry University, Beijing 100083, China; liubowen12156@bjfu.edu.cn (B.L.); bingjiansun@bjfu.edu.cn (B.S.); 2Department of Civil, Construction, and Environmental Engineering, North Dakota State University, Fargo, ND 58102, USA; ying.huang@ndsu.edu

**Keywords:** smoke detection, Raspberry Pi, LBP feature type, cascade classifier

## Abstract

The advances in developing more accurate and fast smoke detection algorithms increase the need for computation in smoke detection, which demands the involvement of personal computers or workstations. Better detection results require a more complex network structure of the smoke detection algorithms and higher hardware configuration, which disqualify them as lightweight portable smoke detection for high detection efficiency. To solve this challenge, this paper designs a lightweight portable remote smoke front-end perception platform based on the Raspberry Pi under Linux operating system. The platform has four modules including a source video input module, a target detection module, a display module, and an alarm module. The training images from the public data sets will be used to train a cascade classifier characterized by Local Binary Pattern (LBP) using the Adaboost algorithm in OpenCV. Then the classifier will be used to detect the smoke target in the following video stream and the detected results will be dynamically displayed in the display module in real-time. If smoke is detected, warning messages will be sent to users by the alarm module in the platform for real-time monitoring and warning on the scene. Case studies showed that the developed system platform has strong robustness under the test datasets with high detection accuracy. As the designed platform is portable without the involvement of a personal computer and can efficiently detect smoke in real-time, it provides a potential affordable lightweight smoke detection option for forest fire monitoring in practice.

## 1. Introduction

Fire is one of the most common and frequent disasters, which may threaten the safety of human lives and properties and destroy the ecological environment and natural resources on which human survival depends [1]. For an effective fire evacuation, it is of great importance to detect the early occurrence of a fire so that the impacted residents and businesses can be alarmed for mitigation actions to protect them against losses of lives and properties [2].

Fire detection methods can be divided into two categories based on the detection of smoke or flame. In a complex environment, flames can be easily blocked in an early stage of a fire, while smoke tends to spread faster and it is easier to be detected. Therefore, smoke detection is an important measure to detect fire in an early stage, which plays an important role in fire monitoring and prevention [3].

A smoke detection system is composed of an acquisition system and a perception system. Most current acquisition systems mainly include a sensor and visual acquisition, among which there are sensors such as gas sensor [4], photoelectric sensor [5], ion sensor, etc. [6], and visual acquisition equipment like camera equipment. Perception systems mainly include a personal computer (PC) or workstation and embedded platform [7]. However, the traditional smoke sensors are mainly used in relatively closed scenes, and they have relatively low stability and are easily affected by external factors such as ambient airflow, detection distance, and thermal barrier effects [8].

With the development of machine vision technology, the vision-based detection system has achieved considerable improvements in detection accuracy for smoke detection. Compared with the sensor-based system, it has the advantages of a wide application range and easy maintenance. Nevertheless, to date, such a vision-based system needs to be connected to a PC or workstation as the perception system requires long-distance circuit deployment which brings difficulties to the layout of the smoke monitoring system in a complex environment. This limitation results in an insufficient coverage rate and low utilization rate of the monitoring system, which is unable to carry out timely and accurate monitoring and warning. Therefore, the development of a smoke detection system based on the visual acquisition and embedded platform perception is of great significance to improve the coverage of the video monitoring system and detection accuracy of the global monitoring system and reduce the false alarm rate e of smoke monitoring.

For the visual perception algorithms, they can be divided into traditional methods [9] and deep learning methods. Due to the complexity of smoke features, deep learning methods used for smoke detection often require many parameters and use a large amount of relevant data for training. As a result, when the embedded platform is selected as the perception system, the storage space required by the deep learning method cannot meet the needs of the lightweight requirements of the platform, and the lack of data sets in the field of smoke detection brings another big challenge to the smoke detection method based on deep learning. Therefore, the smoke detection method based on traditional manual design features can better meet the development requirements and practical application scenarios for smoke detection using an onside platform. However, currently, the focus of smoke detection methods based on traditional hand-designed features lies in the feature design and classifier selection. The accuracy and generalization ability of the method needs to be improved by designing better feature synthesis, in addition to reducing the variation of detection results from different classifiers.

The traditional image- or video-based detection mainly characterizes the static (color, texture, shape, etc.) and dynamic (movement direction, fixed source, smoke fluid properties, etc.) features of smoke by manual extraction [10,11,12,13,14]. For example, by extracting color features in different color spaces (RGB, YCbCr, HSV) [15,16,17,18], ref. [19] compared different color models and merged the HSV and YCrCb spaces to form the final color model. Texture features can be extracted by wavelet decomposition, Gabor transform, and local gradient direction histogram methods. Gao and Cheng [20] proposed the concept of “smoke root” which is defined as a stable smoke source. The “smoke root” does not change in respect of time, which is considered to be the biggest difference between forest fire smoke and disturbance. Since the ViBe algorithm cannot detect long-distance light smoke, the extraction of smoke contours is not complete. In the literature, ref. [21] combines the ViBe algorithm and the MSER algorithm through Bayesian theory to form a better shape of the smoke candidate region, which is used for complete and full-scale video smoke detection. However, a single static feature is usually greatly affected by the environment, and it is difficult to describe the overall characteristics of smoke well. To address this problem, dynamic features from relationships between frames can be considered in combination with static features. Lou and Cheng [22] used a multi-feature fusion method combining dynamic and static stacking strategies to extract smoke regions from continuous video frames and screened out candidate points with high confidence through a multi-frame discrete confidence determination strategy. The candidate points are put into a 2D smoke simulation engine for smoke to generate simulated smoke.

Considering the limited computing power of the embedded platform, most of the above features are complex in design and highly computable, so they are not suitable for the embedded platform. As local binary pattern (LBP) features are fast in computing and can effectively express the texture features of smoke, this feature will be used in the smoke detection algorithm in this paper. In addition, in order to better extract features, the image is often preprocessed before feature extraction, including image binarization, image noise reduction, image enhancement, image geometric transformation, and image interpolation. In this paper, image denoising based on Gaussian filter and image enhancement based on histogram equalization are adopted.

After the smoke area feature extraction, it is necessary to identify and judge the smoke area. At present, the more mature and widely-used classifiers include the support vector machine (SVM), AdaBoost, k-nearest neighbor (KNN), conditional random field (CRF), and Hidden Markov (HMM) [23,24,25,26,27]. Among them, the k-nearest neighbor is often used to detect the effectiveness of features, and SVM and AdaBoost are mostly used for combining different features to improve the final prediction or recognition accuracy due to their high classification efficiency. In addition, the mutual combination of classifiers is also a hot research topic. Cascade classifiers have the advantage of easy training and a lightweight model, so this paper uses cascade classifiers to classify extracted features.

Based on the above-mentioned literature review, in this paper, an embedded smoke front-end sensing system integrated with four modules is designed. A training set from the public database is used to train the cascade classifier based on the LBP feature, and, accordingly, the weight file is obtained followed by transplanting the classifier and its weight into the image processing module. Video frames are obtained from the recorded video and preprocessed by Gaussian filtering and histogram equalization. The smoke is then detected by analyzing the video frames captured by the camera using a trained cascade classifier. If detected smoke exceeds a set threshold, an alarm is issued for early monitoring and warning of smoke. The effectiveness and accuracy of the designed smoke detection system are verified by case studies.

The remaining paper is organized as follows. Section 2 introduces the platform design, including the selection and configuration of the hardware, the module settings of the software system, and the design of the smoke detection algorithm. The effectiveness of the smoke detection algorithm is detailed in Section 3. Finally, Section 4 summarizes the conclusions and identifies potential future work.

## 2. Platform Design

Figure 1 shows the overall platform design. The platform can be divided into four basic modules, including an image input module, an image processing module, a detection and display output module, and an alarm module. The image processing module includes video frame preprocessing and cascade classifier detection based on LBP features, and the output display module includes display output and remote access.

### 2.1. Development Board Platform Selection

This paper compares three popular development board versions: Arduino [28,29,30], BeagleBone Black [31,32], and Raspberry Pi [33,34,35], and the comparison information is shown in Table 1.

On the subject of the development environment, Arduino is simple in design and is developed based on a single-chip microcomputer. It can only run one program at a time, and only supports low-level C++ language, which has obvious limitations. On the other hand, both Raspberry and Beaglebone are based on Linux systems, which can run multiple programs at the same time, support multi-threaded operations, are compatible with multiple programming languages, and have rich development modules. In addition, they can run on the Flash card, and continue to develop new projects by switching to a different or larger memory card, which is convenient for the development of various environments or software.

In terms of operating speed, Arduino’s turnover rate is about 40 times slower than the other two, and the RAM is 1/128,000 of the other two. The Raspberry Pi 4B is equipped with a quad-core processor, large memory space, and the fastest running speed, which is ideal for processing-oriented devices such as camera image processing, video processing, or building an IoT switch that needs to process large amounts of data and send it to the cloud and run artificial intelligence algorithms that require relatively high computing power.

What is particularly outstanding is that the OpenCV module is integrated into the Raspberry Pi, which is convenient for image processing operations, and greatly reduces the difficulty of programming and the complexity of the program. On the other hand, the Raspberry Pi is very friendly to the Python algorithm. The Raspberry Pi comes with the relevant operating environment and compiler, and multiple modules support the operation of Python. With the continuous iterative upgrade of this version, its running speed is getting faster and faster, which provides the basis for real-time smoke detection. In the design of this paper, the running speed, developability, and compatibility with Python of the development board are the important criteria.

After the comparison, the 4B version of the Raspberry Pi is selected in this paper, as shown in Figure 2. It is equipped with a 64-bit quad-core ARMcortex-A72 processor, a Micro SD card as the memory drive, four USB ports, and a Gigabit Ethernet port around the card’s motherboard for peripheral connections. It also has onboard Wi-Fi and Bluetooth modules, a 1.4 GHz main frequency, and 4 GB of running memory for fast computing power to meet the efficiency of data transmission and real-time image processing [36,37].

### 2.2. Hardware Design for Video Input Module

Figure 3 shows the hardware design block diagram. The recorded videos as video sources will be input to the hardware which is mainly composed of the Raspberry Pi microprocessor and the peripherals (display) of the microprocessor. The hardware system is based on the Linux environment. The environment required for OpenCV is configured on the Raspberry Pi and the smoke target detection is completed through the image processing module, and then the remote connection software is performed, with video access or direct output display on the monitor.

### 2.3. Software Design for Image Processing Module

Considering the diversity of images, the infrared images have low contrast, blurred visual effects, and the resolution is not as high as that of traditional visible-light cameras. Due to the influences of imaging conditions, spectral imagery can be coarse and contain a large number of mixed pixels. Although there are some methods that can solve these problems partially, for example, ref. [38] proposes an SRM algorithm based on space-spectral correlation, which uses spectral images to directly extract the utilized spectral characteristics, avoiding spectral unmixing errors, and obtains better mapping results than existing methods. However, the real-time performance of smoke detection will be reduced accordingly. While RGB color space is the most basic, most commonly used, hardware-oriented color space in image processing, and it is easy to obtain and carry out subsequent processing, this paper selects RGB image as the research object.

#### 2.3.1. Denoising Process

Video images usually contain varying degrees of noise, which can be caused by a variety of factors, including recording equipment, the external environment, and data transmission. On the one hand, the presence of noise makes the information of the image more redundant and the data more computationally intensive, and, on the other hand, it can reduce the detection accuracy of the learning model, such as an increase in false alarm rate. Therefore, the image needs to be denoised before input detection, and the noise reduction methods commonly used in the field of target detection are mean filtering, median filtering, and Gaussian filtering. Using the above filtering methods, the results are shown in Figure 4.

In terms of parameter settings for filtering, both the mean and median filters were processed using a 5 × 5 sub-region, and for consistency, the Gaussian filter used a 5 × 5 convolution kernel. From the above test samples, it shows a loss of the sharpness of images caused by the mean filter and median filter while the Gaussian filter reduces in turn. Moreover, the first two have a stronger sense of smearing on the screen, which may mistakenly confuse the light-colored area in the background with the lighter concentration of smoke in the foreground during detection, resulting in false detection or missed detection. In summary, this paper selects a Gaussian filter with a convolution kernel of 5 × 5 for the image denoising process.

#### 2.3.2. Histogram Equalization

In this paper, the texture feature of smoke is also used as a smoke identifier. The Local Binary Pattern (LBP) operator using histogram equalization is applied to extract the smoke features, which are processed based on the pixels of the region. The histogram equalization can evenly map the gray level of the original image to the entire gray level range to obtain an image with uniform gray level distribution, thereby enhancing the overall contrast effect of the image and making the details of the picture clearer. In this way, smoke areas and non-smoke areas can be effectively distinguished when calculating features. The greater the difference between the feature values of smoke and other regions during feature extraction, the more beneficial the histogram equalization is to improve the accuracy of detection. Figure 5 shows the pixel changes before and after histogram equalization and the process of applying the histogram equalization. Figure 6a,b compare the images before and after the histogram equalization. It is clearly indicated in Figure 6a,b that the distinction between pixels is significantly enhanced and the smoke in the image is much clear after the histogram equalization.

#### 2.3.3. Cascading Classifier Based on LBP Features

(1)LBP features

In smoke detection, texture features are often used as a feasible method to characterize smoke. As an effective texture description operator, the LBP operator has remarkable characteristics such as rotation invariance and grayscale invariance, and, at the same time, eliminates the problem of illumination changes to a certain extent. As shown in Figure 7, for the original LBP operator, in a 3 × 3 window, the pixel value of the center point in the area is the threshold value, and the gray value of the adjacent eight-pixel points is compared one by one. If the surrounding points are smaller than the center point, it becomes 0, otherwise 1. The LBP operator then arranges the 0 and 1 values obtained by each pixel point in a certain order and converts them into the corresponding decimal number, that is, the pixel value of the current center point. Specifically, the basic LBP operator is defined as:(1)$$LBP\left(P,R\right)=\sum_{Q=0}^{P-1}S\left(g_{q}-g_{c}\right)\cdot 2^{p}$$
(2)$$S\left(x\right)=\{\begin{array}{l} 1\;\;x\geq 0\\ 0\;\;x<0 \end{array}$$
in which P is the number of sampling points, R is the sampling radius, $g_{c}$ is the gray value of the central pixel in the local area, and $\;g_{q}$ is the gray value of the *q*th sampling point in the neighborhood of the central pixel.

An LBP operator can generate a coding mode of a combination of 0 and 1. With the continuous increase of sampling points, the calculation amount will continue to increase, and at the same time, it will bring redundant information. To solve such a problem and improve the efficiency of the operator, Ojala proposed to use an equivalent model to reduce the dimension of the model types of the LBP operator, and the number of models was reduced from the original 2*p* to *p* (*p* − 1) + 3 [39]. The LBP mode value of the equivalent mode can be defined as follows:(3)$$LBP_{P,R}=\{\begin{array}{c} \overset{P-1}{\underset{Q=0}{\sum }}S\left(g_{q}-g_{c}\right)\cdot 2^{p}\;\;U\left(LBP_{P,R}\right)\leq 2\\ P+1\;\;\;\;\;\;\;\;\;\;\;\;\;\;\;otherwise \end{array}$$
where $U\left(LBP_{P,R}\right)\;$ indicates the number of transitions between 0 and 1 in the basic LBP mode. To convert LBP features into usable information, it is necessary to divide the LBP feature image into m local blocks, extract the histogram of each local block, and then connect these histograms in turn to form the statistical histogram of LBP features, that is, LBPH, and finally use the machine learning method to train the LBP feature vector for image detection.

In the process of target detection, it is difficult for a single classifier to achieve a high detection accuracy. Therefore, this paper selects multiple strong classifiers trained by the Adaboost algorithm to form a cascade classifier with higher detection efficiency to better achieve smoke detection. The initial weak classifier adopts a small number of feature dimensions, and the judgment result has great randomness, so it is necessary to train the optimal weak classifier before forming a strong classifier in parallel. For each feature, the weak classifier determines the best threshold classification function, the mathematical structure is as follows:(4)$$h\left(x,f,p,\theta \right)=\{\begin{array}{c} \;1\;\;\;\;\;\;pf\left(x\right)<p\theta \\ \;\;\;\;0\;\;\;\;\;\;\;otherwise\;\;\;\; \end{array}$$
where x is the size of the sub-window, f represents the feature, θ is the threshold, and p represents the polarity to indicate the direction of the inequality.

(2)Adaboost strong classifier training

As shown in Figure 8, the strong classifier is composed of multiple weak classifiers in parallel and is generated after T rounds of iteration. Finally, multiple strong classifiers are cascaded from simple to complex, and the accuracy rate is improved through continuous training. Each strong classifier has better performance in the cascade classifier chain. The detailed training process of the cascade classifier is listed in Algorithm 1.
**Algorithm 1 The Detailed Cascade Classifier Training Steps**(1.)Set the minimum detection accuracy which is required to be achieved for each layer $d_{min}$, maximum false positive rate $f_{max}$, and the false recognition rate of the final cascaded classifier $F_{t}$;(2.)Obtain the sample set required for training, and perform preprocessing such as normalization, noise reduction, etc.(3.)Initialization: $F_{0}=1$, $D_{0}=1$
(4.)i=0;  for: $F_{i}>F_{t}$    ++i;    $n_{i}=0$;    $F_{i}=F_{i-1}$;  for: $F_{i}>f\times F_{i-1}$    $++n_{i}$;    Using AdaBoost to train a strong classifier with $n_{i}$ weak classifiers on the training set;     
Measure the detection rate $D_{i}$ and false recognition rate of the current cascade classifier $F_{i}$, satisfy $F_{i}\leq f_{max},\;D_{i}\geq d_{min}$;for:$d_{i}>d\times D_{i-1}$        ·Lower the strong classifier threshold for layer i;        ·Measure the detection accuracy $D_{i}$ and false recognition rate of the current cascade classifier $F_{i}$;      ·The number of negative samples N remains the same in the update;      ·Use the current cascade classifier to detect non-face images, and put the misidentified images into N;(5.)Save the training results.

The resolution of the samples during cascade classifier training is usually smaller than that of the detection samples, thus, multi-scale detection is required during the smoke detection. There are two general mechanisms for multi-scale detection. The first one is to continuously scale the image without changing the size of the search window. This method requires the operation of regional feature values for each scaled image, which is inefficient. The other method is to continuously initialize the search window size to the image size during training and continuously expand the search window to search, which addresses the shortcomings of the first method. The cascade classifier used in this paper adopts the second strategy by (1) searching, detecting, and calling the detected multi-scale function in OpenCV, (2) traversing the image through the initial window size of 35 × 35, (3) calculating the characteristics of the object and comparing them, and (4) gradually enlarging the length and width of the detection window according to a certain proportion. By repeating the above operation, the detection target position and frame selection are determined.

### 2.4. Email Alarm Module

As shown in Figure 9, when the system detects the presence of smoke, the imwrite function of OpenCV is used to automatically save five frames of pictures when it is determined that smoke is detected and generate a file attachment. It uses the socket to establish a link with the server and sends the SMTP commands. The system transmits the content of the mail and the SMTP commands between the remote end and the server in the form of text streams and attachments to achieve mailbox alarms.

### 2.5. Display Output Module

As shown in Figure 10, the detected smoke information can be obtained and displayed using either direct or remote access. For direct access, the external display is directly connected to the Raspberry Pi motherboard with an HDMI cable, using a mobile power supply through the USB Type-c power port to the Raspberry Pi motherboard for a power supply to achieve the real-time output of detection. For the remote access, the remote PC can be connected to the Raspberry Pi using a VNC service by obtaining the IP address of the Raspberry Pi and logging into the platform remotely to achieve a remote signal and data transmission.

## 3. Case Studies and Discussion

The datasets used in the training process of the platform in this study are obtained from three open-source smoke sample sets, including smoke videos and pictures from:(1)Laboratory of Bilkent University, Turkey [40](2)Yuan Feiniu Laboratory, Jiangxi University of Finance and Economics [41](3)CVPR Laboratory of Keimyung University, Korea [42]

The training sample set includes 2800 positive samples and 4000 negative samples. As shown in Figure 11 and Figure 12, the grayscale images of some positive and negative training samples are selected, respectively. The resolution of positive samples is 35 × 35, and for the negative samples it is 48 × 48.

After the cascade training, to validate the developed platform, eight test videos are selected to test the trained cascade classifier, respectively. To verify the robustness and effectiveness of the algorithms, the selection of the eight groups of videos takes into account the diversity of the background, the intensity of the light, and the distance. Table 2 lists the descriptions of the eight groups of test videos.

### 3.1. Experimental Indicators

To evaluate the developed platform based on the experimental results from the case studies, the confusion matrix is used to count the actual smoke in the three cases as a reference as shown in Table 3. In Table 3, TP represents the true smoke, when the number of frames where the smoke frame in the test video is identified as the smoke frame, FP is the false positive, when the number of frames where the smoke frames in the test video are identified as non-smoke frames, TN is the true negative, when the number of frames in the test video where non-smoke frames are identified as non-smoke frames, and FN is the false negative, the number of video frames where non-smoke frames in the test video are identified as smoke frames.

After filling out the statistical counting form based on the testing data of the case studies, it can be used to calculate the corresponding experimental indicators. In this paper, the accuracy, detection rate, false alarm rate, and missed detection rate are selected as the main experimental indicators as they are commonly used in the field of smoke detection. The calculation of these indicators is further explained as follows:

Accuracy (ACC): The ratio of the number of frames in the test video for which all predictions are correct (smoke frames are predicted as smoke frames, non-smoke frames are predicted as non-smoke frames) in the total number of video frames, which can be represented as:(5)$$\mathrm{ACC}=\frac{\mathrm{TP}+\mathrm{TN}}{\mathrm{TP}+\mathrm{FN}+\mathrm{FP}+\mathrm{TN}}$$

True positive rate (TPR): The proportion of the smoke frames in the test video that are correctly detected as smoke frames in all smoke frames can be represented as:(6)$$\mathrm{TPR}=\frac{\mathrm{TP}}{\mathrm{TP}+\mathrm{FN}}$$

False-positive rate (FPR): The ratio of the number of non-smoke frames in the test video that are falsely detected as smoke frames in all non-smoke frames can be represented as:(7)$$\mathrm{FPR}=\frac{\mathrm{FP}}{\mathrm{FP}+\mathrm{TN}}.$$

False-negative rate (FNR): The ratio of the number of smoke frames that are not detected in the test video to all smoke frames can be represented as:(8)$$\mathrm{FNR}=\frac{\mathrm{FN}}{\mathrm{TP}+\mathrm{FN}}.$$

In some test videos, smoke will be included in the whole process. At this time, since TN and FP do not exist, ACC = TPR, and there is no FPR. In addition, the speed or consumed time is also investigated using the measurement speed in frames per second (FPS) to investigate the efficiency of the developed platform.

### 3.2. Test Result Discussion

The used training computer has a configuration of Intel-i7 processor, NVIDIA GTX1080 GPU, and 16G memory. Table 4, Table 5 and Table 6 show the specific training parameter settings used in the case studies.

Discrete AdaBoost (DAB) and Gentle AdaBoost (GAB) are currently the most commonly used AdaBoost algorithms. Among them, the GAB has evolved based on DAB. To explore the feasibility and effectiveness of the Adaboost algorithm, by setting the classification of the enhanced classifier, the training time can be compared between the DAB, which is the traditional classifier, and the GAB cascade classifier used in the developed platform in this paper under the same conditions, and the detection indicators of the three sets of test videos can be identified and compared. The time used for the DAB cascade classifier is 64 min 38 s, and the time used by the GAB cascade classifier is 45 min 50 s. Table 6 shows the results from the validation tests of the three case studies.

To improve the real-time detection of video, the resolution of video frames can be reduced by down sampling. However, due to the slow change of foreground and background between consecutive frames, the detection can be performed by inter-frame extraction. The former will lose the accuracy of the image when processing and may affect the accuracy of detection, so this paper takes the latter approach for video testing. Table 7 shows the results from the validation tests of the eight case studies.

It can be found from Table 7 that after the test of the eight groups of videos, the average detection speed (FPS in Table 7) of the enhanced classifier based on the GAB algorithm is about 11 frames faster than that of the enhanced classifier based on the DAB algorithm. So the detection speed of the former is significantly higher than that of the latter. The distance of smoke in the frame differs in videos 3 and 8, and video 7 has a higher frame brightness. The test results show that the classifier based on the GAB algorithm is less affected by the distance, and the brightness of the frame has not been greatly affected for both classifiers. On all non-smoke frames, the former has a better detection effect. However, in test video 2, the GAB has a high false alarm rate, resulting in a low accuracy rate. The possible reason is that the smoke concentration in the early test video 2 is low and the distribution area is small because the DAB requires weak classification. The output of the detector is binary, and the GAB will relax the requirements to real values. When performing multi-scale sliding window detection on these test frames, the former will output positive samples as long as smoke is detected, while the latter needs to output close to negative samples. The label value of the sample is more likely to be determined as a negative sample by the classifier threshold, resulting in a higher false-positive rate.

From Table 7, it also can be seen that compared to the traditional DAB, the GAB cascade classifier not only has a shorter training time under the same training conditions but also has a faster calculation speed, which can meet the lightweight needs of the algorithm and has a better average performance on the different test sets. The video frames with large differences in the shape distribution of smoke in the three test samples are intercepted, and the detection result of the GAB-based cascade classifier is shown in Figure 13.

## 4. Conclusions and Future Work

This paper develops a lightweight portable real-time smoke front-end sensing device based on Raspberry Pi, which can effectively reduce the cost of smoke detection and miniaturize the smoke detection device. Specifically, the following conclusions can be drawn:(1)The Adaboost cascade classifier based on LBP features is applied in the developed platform to ensure the detection accuracy and real-time performance of the method.(2)With the iterative upgrade of the Raspberry Pi version, the configuration of hardware and software has been greatly improved to meet the needs of processing a large number of low-level image features extracted by the lightweight learning model.(3)The characteristics of Raspberry Pi, such as the small size, strong computing power, and abundant peripherals, provide a good platform for hardware development of smoke detection, which can be applied to more complex and changeable environments.(4)The GAB-based cascade classifier has better performance in training and testing, including shorter training time, high detection efficiency, and low missed detection rate, but the detection effect when the smoke concentration is low and the shape distribution is discrete could be improved.

Due to the use of the embedded platform, the developed platform in this paper has the advantages of simple operation and flexible expansion, which can meet the diversification of needs in many field applications which requires portable and lightweight smoke detection devices. Limited to the memory and computing power of the Raspberry Pi, the embedded learning model and the features extracted by the network need to be lightweight and simplified. Although the cascade classifier used in this paper can match the configuration of the Raspberry Pi, the false alarm rate is still relatively high, and there is a certain deviation in the positioning of the smoke area. Therefore, the effectiveness, diversity, and lightweight of the algorithm to extract smoke features are the key problems to be solved by the future development platform.

## Figures and Tables

**Figure 1 sensors-22-04655-f001:**
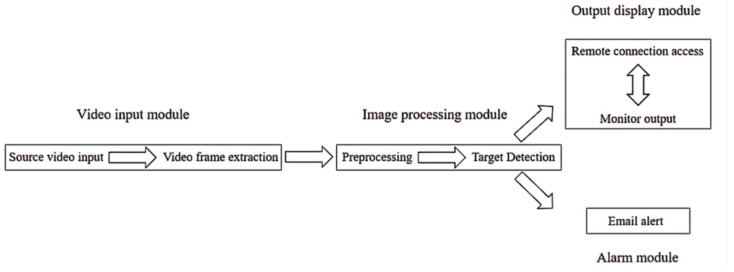
System overall design block diagram.

**Figure 2 sensors-22-04655-f002:**
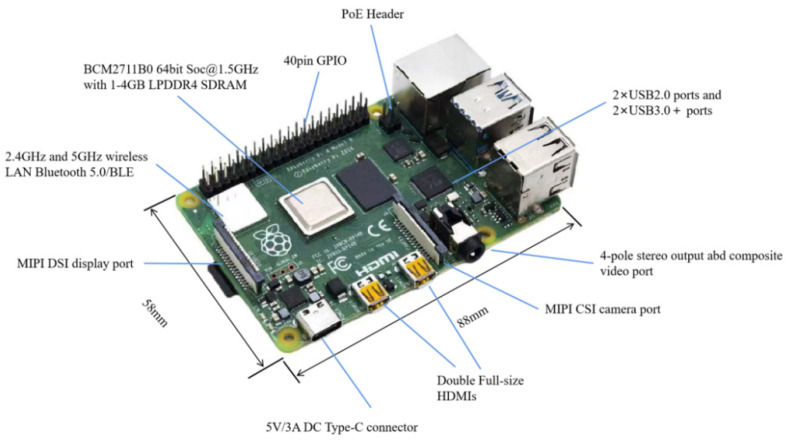
Raspberry Pi 4B Microprocessor.

**Figure 3 sensors-22-04655-f003:**
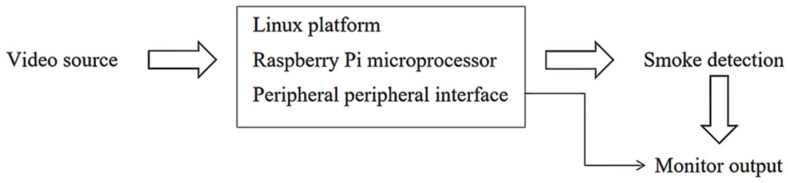
Hardware Design Block Diagram.

**Figure 4 sensors-22-04655-f004:**
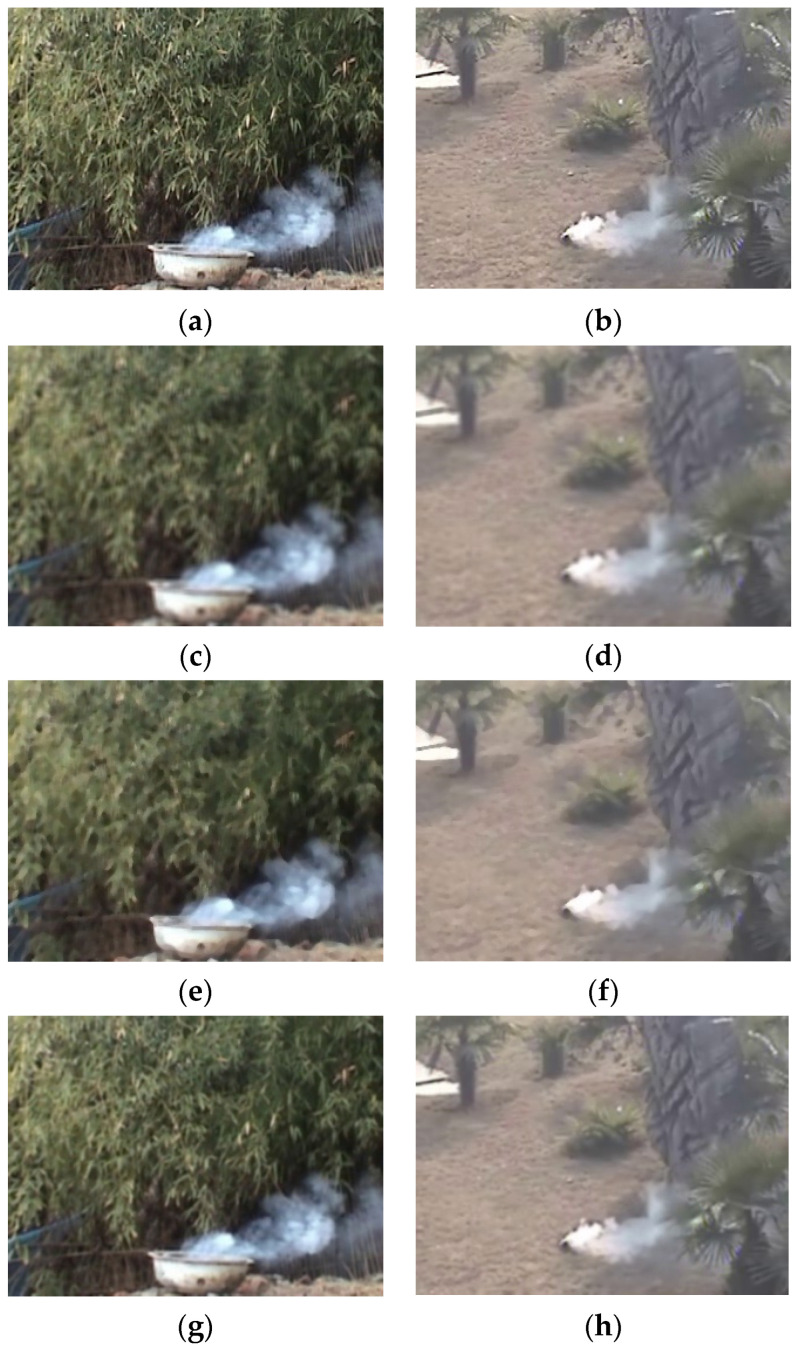
Example of filtering effect: (**a**,**b**): original image; (**c**,**d**): mean filtering; (**e**,**f**): median filtering; (**g**,**h**): Gaussian filtering.

**Figure 5 sensors-22-04655-f005:**
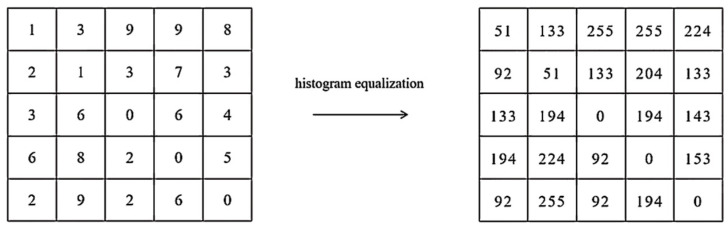
Histogram equalization of pixels.

**Figure 6 sensors-22-04655-f006:**
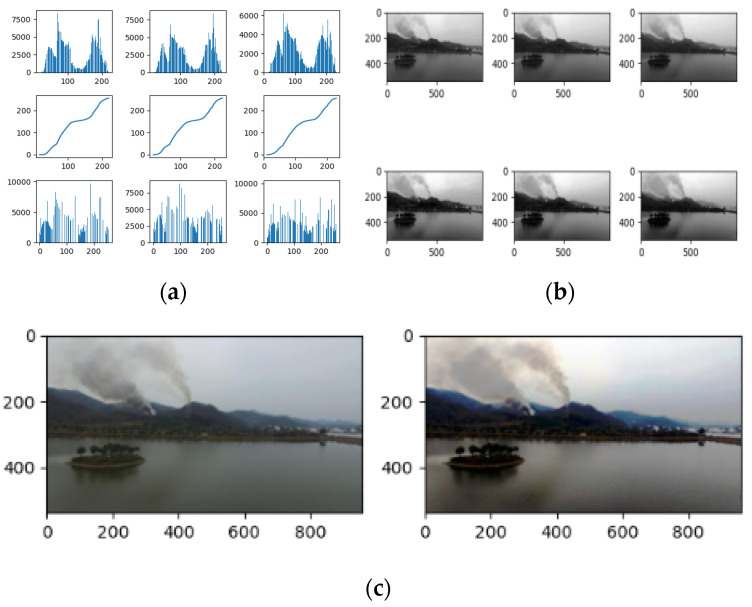
Histogram equalization: (**a**) original histogram of RGB channel, equalized histogram; (**b**) original image of RGB channel, equalized image; (**c**) original image, equalized image.

**Figure 7 sensors-22-04655-f007:**
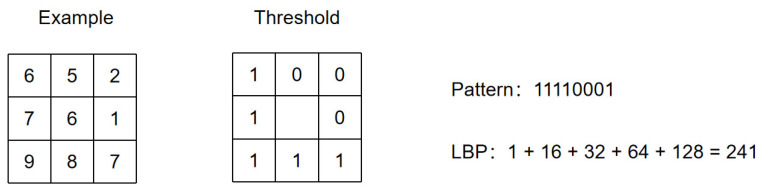
Original LBP operator.

**Figure 8 sensors-22-04655-f008:**
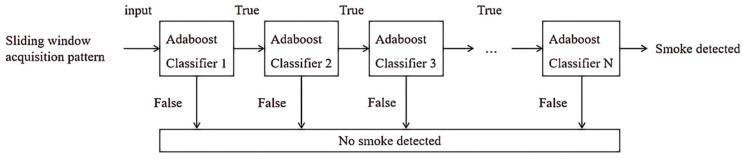
Cascade classifier structure diagram.

**Figure 9 sensors-22-04655-f009:**
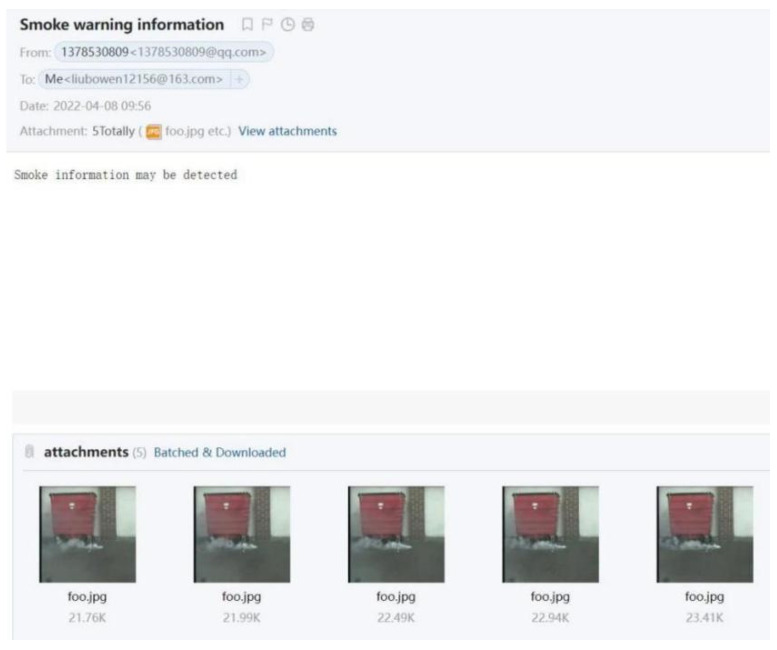
Email alarm interface.

**Figure 10 sensors-22-04655-f010:**
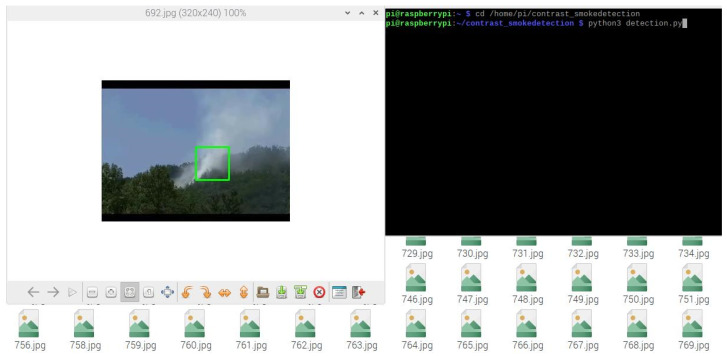
Remote connection interface.

**Figure 11 sensors-22-04655-f011:**
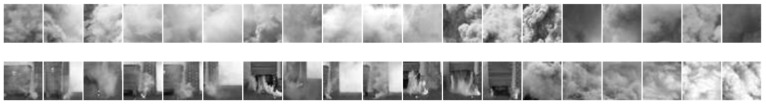
Positive sample examples.

**Figure 12 sensors-22-04655-f012:**
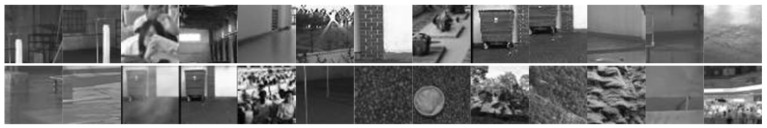
Negative sample examples.

**Figure 13 sensors-22-04655-f013:**
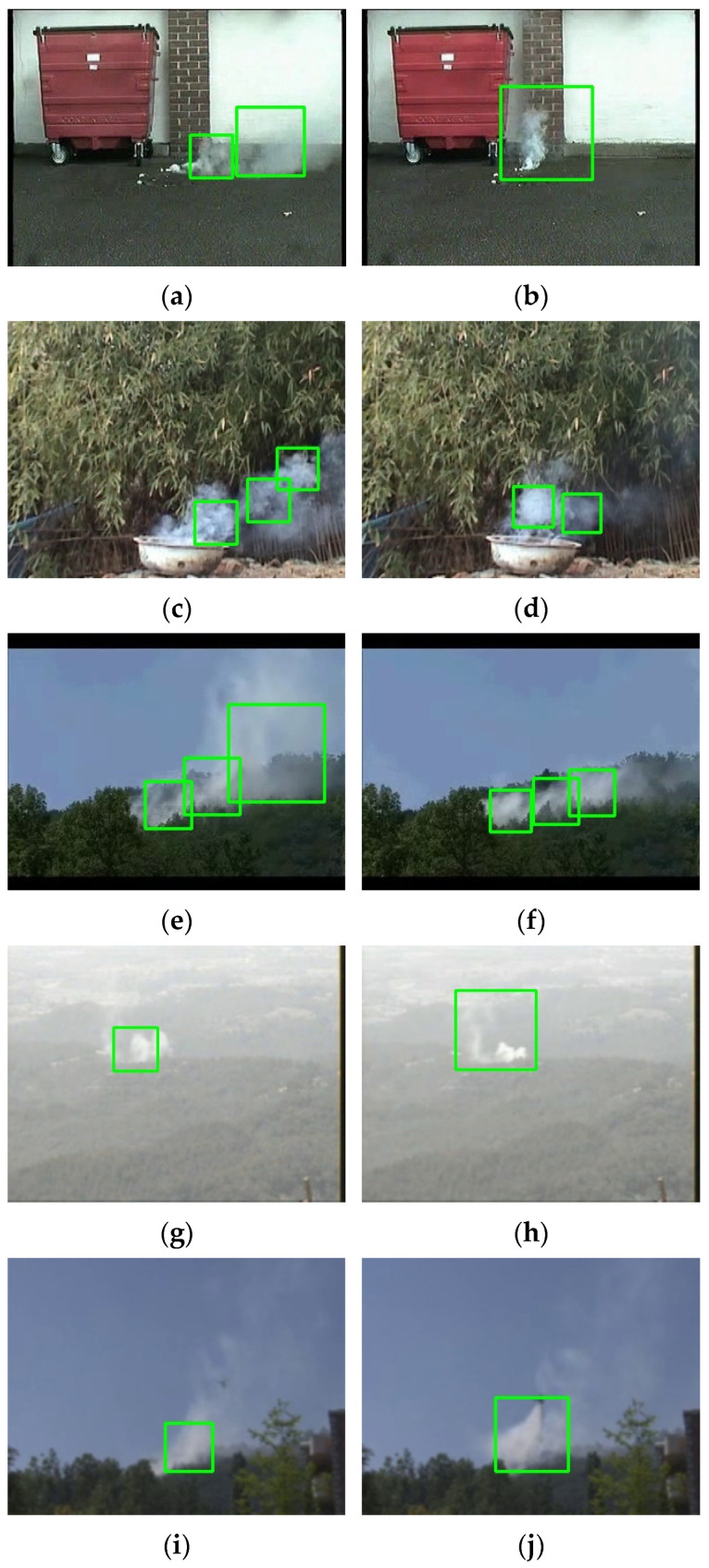
Example of detection effect where regions in green boxes are detected as smoke region: (**a**) 307th frame of test video 1 (**b**) 674th frame of test video 1 (**c**) 237th frame of test video 2 (**d**) 778th frame of test video 2 (**e**) 807th frame of test video 3 (**f**) 1414th frame of test video 3 (**g**) 454th frame of test video 7 (**h**) 584th frame of test video 7 (**i**) 698th frame of test video 8 (**j**) 770th frame of test video 8.

**Table 1 sensors-22-04655-t001:** Development board parameters comparison.

Development Board	Arduino Uno	Raspberry Pi	BeagleBone Black
Version	R3	4B	Rev C
Price	$20	$150	$100
Size	53 × 68	58 × 88	53 × 86
Processor	ATmega328P	ARMcortex-A72	AM335x
Clock frequency	16 MHz	1.4 GHz	1 GHz
RAM	2 KB	1/2/4 GB	512 MB
Storage memory	32 KB	(SD Card)	4 GB (microSD)
Input voltage	7–12 V	5 V	5 V
Minimum power	20 mA (0.14 W)	3 A (15 W)	350 mA (1.75 W)
GPIO	14	40	88
Analog input	6–10 bit	N/A	7–12 bit
UART	1	1	5
Dev IDE	Arduino Tool	Python /Scratch/Squeak/Linux	Python/Scratch/Squeak/Linux/Cloud9
USB Master	N/A	2 USB2.0, 2 USB3.0	2 USB2.0
video output	N/A	HDMI, Composite	N/A
Audio output	N/A	HDMI, Analog	Analog

**Table 2 sensors-22-04655-t002:** Test video description.

Test Video	Resolution	Total Frames	Smoke Frames	Illustration
video test1	320 × 240	900	835	A smoking device is thrown to the ground and releases white smoke, smoke movement is chaotic, and there are distractors such as spilled water in the middle.
video test2	320 × 240	1200	1050	Simulate forest fire smoke, and create smoke among leaves. Smoke moves roughly up and to the right, and the background leaves frequently shake.
video test3	320 × 240	2326	2326	In mountainous areas, the direction of smoke movement is to the upper right and the distance is relatively short. The picture is dark.
video test4	320 × 240	4536	0	Pedestrians in the background, no smoke
video test5	320 × 240	1000	0	Cars on the road, no smoke
video test6	320 × 240	894	0	Large areas of continuously shaking leaves
video test7	320 × 240	606	606	Mountainous areas, smoke is far away and the brightness of the picture is high.
video test8	320 × 240	2886	2886	Mountainous area, the direction of smoke movement is more confusing, and the picture has moving objects such as firefighting fliers, far away.

**Table 3 sensors-22-04655-t003:** Confusion matrix.

Smoke Classification		Actual Situation	
		Smoke	Non-Smoke
Predicted results	Smoke	TP	FP
	Non-Smoke	FN	TN

**Table 4 sensors-22-04655-t004:** General parameter settings.

Number of Positive Samples	Number of Negative Samples	Number of Stages	PrecalcValBufSize	PrecalcIdxBufSize
2800	4000	20	1024 MB	1024 MB

**Table 5 sensors-22-04655-t005:** Gradient parameter settings.

Stage_Type	Feature_Type	Width	Height
BOOST	LBP	35	35

**Table 6 sensors-22-04655-t006:** Enhanced classifier parameter settings.

Boosted Type	Min_Hit_Rate	Max_False_Alarm_Rate	Weight_Trim_Rate	Max_Depth	Max_Weak_Count
GAB/DAB	0.999	0.5	0.95	1	100

**Table 7 sensors-22-04655-t007:** Based on test videos test results.

Test Video	Adaboost Type	FPS	Test Results							
			TP	TN	FN	FP	ACC (%)	TPR (%)	FPR (%)	FNR (%)
video_test1	LBP + GAB	28.52	410	33	6	1	98.44	98.56	2.94	1.44
	LBP + DAB	17.81	279	31	137	3	68.89	67.07	8.82	32.93
video_test2	LBP + GAB	25.80	365	84	150	1	74.83	70.87	1.18	29.13
	LBP + DAB	12.65	486	67	29	18	92.17	94.37	21.18	5.63
video_test3	LBP + GAB	23.30	1150	\	13	\	98.88	98.88	\	1.12
	LBP + DAB	12.00	1163	\	0	\	100	100	\	0
video_test4	LBP + GAB	29.68	\	2244	\	24	98.94	\	1.06	\
	LBP + DAB	14.96	\	1831	\	437	80.73	\	19.27	\
video_test5	LBP + GAB	28.34	\	438	\	62	87.60	\	12.40	\
	LBP + DAB	13.42	\	300	\	200	60.00	\	40.00	\
video_test6	LBP + GAB	26.72	\	398	\	49	89.04	\	10.96	\
	LBP + DAB	20.70	\	337	\	110	75.39	\	24.61	\
video_test7	LBP + GAB	22.34	300	\	3	\	99.01	99.01	\	0.99
	LBP + DAB	11.50	303	\	0	\	100	100	\	0
video_test8	LBP + GAB	27.24	1438	\	5	\	99.65	99.65	\	0.35
	LBP + DAB	15.79	1337	\	106	\	92.65	92.65	\	7.35

## Data Availability

Not applicable.

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
