# Peer review of "An Embedded Portable Lightweight Platform for Real-Time Early Smoke Detection"

_sensors, 2022, doi:10.3390/s22124655_

Round 1

Reviewer 1 Report

This paper presents a light weight portable remote smoke front-end perception platform based on Raspberry pi. however, there are some issues that restrain the quality of the manuscript.

11.       The paper is a bit disorganized and difficult to read. You may reorganize the content of your manuscript, for example, section 1 introduction, section 2 proposed system, section 3 experimental results and section 4 conclusion.

22.       Please write proper algorithm for table 1.

33.       How practical it is to use raspberry pi for real-time detection given that it’s low storage memory and low RAM?

44.   There is a lot of spelling errors. the authors are advised to check and carefully rewrite the manuscript. for instance line 22 “flatform”, line 23 “systemplatform”

55.     What makes your paper different from previously done smoke detector with Raspberry pi?

Reviewer 2 Report

This paper designs a lightweight portable remote smoke front-end perception platform based on Raspberry Pi under Linux operating system. Experimental results show the good performance of the proposed method. However, some issues should be addressed.

Major issues:

1) But I want to know what is the basis for the author to select the three groups of data? Is the smoke detection accuracy of the system different in different environments? For example, when the light is dim or the weather is bad. These can be analyzed and discussed in the experiment.

2) In addition, it seems that the video and images processed by the author are obtained from ordinary optical imaging. Can you also identify infrared, spectral and other types of image information?

Minor issues:

1) The methods proposed by the authors is mainly based on image processing. In the introduction part, some image detection techniques should be further introduced, e.g.,

[1] Super-Resolution Mapping Based on Spatial-Spectral Correlation for Spectral Imagery [J]. IEEE Transactions on Geoscience and Remote Sensing, 2021, 59(3): 2256-2268.

[2] A Simple Method of Mapping Landslides Runout Zones Considering Kinematic Uncertainties, Remote Sensing, 2022, 14(3): 668.

2) In addition, there are some grammatical errors in the article, which need further careful proofreading.

Round 2

Reviewer 2 Report

Thank you for your detailed reply. There is no major problem here. The authors are suggested to further standardize the format of references, for example, some articles have incorrect year marks. Reference [36] does not contain the authors' names.

This manuscript is a resubmission of an earlier submission. The following is a list of the peer review reports and author responses from that submission.